# Shuttling a single charge across a one-dimensional array of silicon quantum dots

A.R. Mills[1], D.M. Zajac[1], M.J. Gullans[1], F.J. Schupp[1], T.M. Hazard[1] & J.R. Petta [1]

Significant advances have been made towards fault-tolerant operation of silicon spin qubits, with single qubit fidelities exceeding 99.9%, several demonstrations of two-qubit gates based on exchange coupling, and the achievement of coherent single spin-photon coupling. Coupling arbitrary pairs of spatially separated qubits in a quantum register poses a significant challenge as most qubit systems are constrained to two dimensions with nearest neighbor connectivity. For spins in silicon, new methods for quantum state transfer should be developed to achieve connectivity beyond nearest-neighbor exchange. Here we demonstrate shuttling of a single electron across a linear array of nine series-coupled silicon quantum dots in ~50 ns via a series of pairwise interdot charge transfers. By constructing more complex pulse sequences we perform parallel shuttling of two and three electrons at a time through the array. These experiments demonstrate a scalable approach to physically transporting single electrons across large silicon quantum dot arrays.

[1] Department of Physics, Princeton University, Princeton, NJ 08544, USA. Correspondence and requests for materials should be addressed to J.R.P. (email: petta@princeton.edu)

Single spin qubits in quantum dots can be fabricated with high areal densities in Si due to their small ~30 nm size[1,2]. In general, electron spins in semiconductors can have spin lifetimes $T_1$ that approach 1 min[3] and coherence times $T_2$ that exceed one second[4]. With single-qubit control fidelities that are competitive with superconducting qubits and trapped ions[5–7], and the first realization of high fidelity two-qubit gates[8–11], it is becoming increasingly important to now direct attention towards the development of a large-scale and highly interconnected spin-qubit architecture[12–16]. Spin qubits in quantum dots are coupled through the exchange interaction at ~50 nm length scales[8–10,17,18]. Spin–photon coupling was proposed[19–22] as a method for interactions over cm length scales and recently the first experimental advances towards a photonic interconnect have been made[23,24]. However, the large footprint of the superconducting cavities required for spin–photon coupling motivates the development of intermediate-scale quantum state transfer (QST) protocols that are effective at 50 nm–10 µm length scales.

There are many theoretical proposals for achieving intermediate-scale QST in quantum dots. Early work suggested coherent transport by adiabatic passage[25] or the implementation of an exchange coupled spin-bus[26]. Charges can also be transported in the moving potential of a surface acoustic wave[27,28], through direct shuttling in a gate-voltage induced traveling wave[12], or by pairwise interdot charge transfers down an array of quantum dots in bucket brigade fashion. The bucket brigade approach has been demonstrated in small GaAs quantum dot arrays[29–32]. However, there are several challenges associated with scaling up the bucket brigade approach. First, for QST of spins, the spin must be transferred on a timescale that is significantly shorter than the inhomogeneous spin dephasing time $T_2^*$. Second, to allow for adiabatic charge transfer, there must be a substantial 1–5 GHz nearest-neighbor tunnel coupling between all dots in the array. Finally, the electron transfer process requires a detailed understanding of multidimensional charge stability spaces[33] and the ability to precisely navigate through these spaces on nanosecond timescales.

Here we demonstrate single charge shuttling through a linear array consisting of 9 Si quantum dots in ~50 ns, more than three orders of magnitude faster than $T_2^*$ ~100 µs in isotopically enriched silicon[34]. We also note that spin dephasing due to hyperfine coupling in natural silicon may be suppressed by motional narrowing during the shuttling process, making the shuttling approach applicable to a variety of host materials (e.g., GaAs, InAs, and InSb)[32]. Our approach for traversing the high dimensional charge stability space can be extended to larger 1D arrays, and possibly two-dimensional (2D) arrays[35], providing a path towards intermediate-scale QST in silicon.

## Results
**Overview.** The experiment is performed using quantum dots defined in an undoped $^{28}$Si/SiGe heterostructure. Lateral confinement of electrons is achieved using a gate design with a repeating unit cell structure consisting of 3 quantum dots and a charge sensor[1]. Large 1D quantum dot arrays can be fabricated by repeating the unit cell. Our device is shown in Fig. 1a and consists of 3 unit cells (9 dots and 3 charge sensors). Plunger gates (P$_1$, P$_2$, etc.) are used to accumulate few-electron quantum dots and barrier gates (B$_1$, B$_2$, etc.) set the tunnel coupling between the dots. Figure 1b shows a cross-sectional scanning electron microscope SEM image of a gate pattern that is similar to the one used in this experiment. The overlapping nature of the Al gate-electrodes, where the Al layers are electrically isolated by a native oxide barrier, results in a high degree of control over the local

electric potential and minimizes capacitive cross-coupling in the device[1].

The charge shuttling sequence for the 9-dot array is illustrated in Fig. 1c, where the quantum dot confinement potential $V(x)$ is modulated in time by applying voltage pulses to the plunger gates on the device. Starting with an empty array of dots, we load one electron onto dot 1 by lowering its chemical potential below the Fermi level of the source reservoir. The electron is then transferred to dot 2 by lowering its chemical potential while simultaneously increasing the chemical potential of dot 1. We repeat the process of pairwise charge transfers (dot 2 → dot 3, dot 3 → dot 4, etc.) until the electron resides in dot 9. The charge shuttling sequence is completed by raising the chemical potential of dot 9 above the Fermi level of the drain reservoir, which unloads the electron from the array. In the absence of shuttling errors, each shuttling cycle will transfer a single electron across the device. Repeating the shuttling process at frequency $f$ will therefore result in a current $I = ef$ through the device.

**Virtual gates.** A high degree of control of charge states in semiconductor double quantum dots (DQDs) has been achieved, as the 2D charge stability diagram that maps out the number of electrons in the left and right dots as a function of the left and right dot gate voltages can easily be measured and visualized[33]. For the 9-dot linear array, it is not feasible to independently control the electronic occupation of each dot in the array using just two gate voltages. Instead, control over the charge states requires the traversal of a 9D gate voltage parameter space spanned by $V_{P1}, V_{P2}, ..., V_{P9}$. To simplify the charge shuttling process, we measure the capacitance matrix of the device and use this knowledge to establish virtual gates which allow for independent control of the chemical potential of each dot in the array (Supplementary Discussion). Through software, the virtual gates largely eliminate the effects of capacitive cross-coupling that would, for example, result in a shift of the dot 2 chemical potential when neighboring plunger gate voltages $V_{P1}$ or $V_{P3}$ are varied. Similar calibrations to reduce the effects of cross-capacitance were utilized in an early multi-junction charge pump experiment[36], GaAs DQDs[37], and a quantum dot Fermi-Hubbard model simulator[38]. In addition, we break the charge shuttling process down into a sequence of pairwise interdot charge transitions that are executed in virtual gate voltage space. We now describe how the virtual gates are established and utilized to implement charge shuttling through the 9-dot array.

The chemical potential $\boldsymbol{\mu}$ of the dots is controlled by changing the plunger gate voltages $\mathbf{V}_P$[33,38,39]. The conversion between $\mathbf{V}_P$ and $\boldsymbol{\mu}$ is determined by a dimensionless matrix $\mathbf{G}$ related to the capacitance matrix for the device and the experimentally measured dimensionless lever arm for dot 1, $\alpha_1 \approx 0.12$, via the formula $= e\alpha_1 \mathbf{G} \mathbf{V}_P$ (see Fig. 2a and Supplementary Discussion). Virtual gate voltages, defined here as $u_i$, effectively invert $\mathbf{G}$, such that a change in $u_i$ only affects the chemical potential of the $i^{th}$ dot, $\mu_i$ (see Fig. 2b). Figure 2c, d illustrate the transition from voltage space to virtual gate voltage space. Figure 2c shows the charge stability diagram of a DQD that is formed by accumulating electrons beneath plunger gates P1 and P2 while the rest of the array is fully accumulated to form a channel to the lead. The charge sensor conductance $G_{S1}$ is plotted as a function of the plunger gate voltages $V_{P1}$ and $V_{P2}$, which change the occupancy $(N_1, N_2)$ of the DQD, where $N_i$ is the number of electrons on dot $i$. Due to cross-capacitance in the device, a change in $V_{P1}$ results in a slight change in the chemical potential of dot 2. As a result, the dots 1 and 2 charge transitions in Fig. 2c are sloped. By measuring the capacitance matrix of the DQD, it is possible to correct for the cross-capacitance and transform into virtual gate

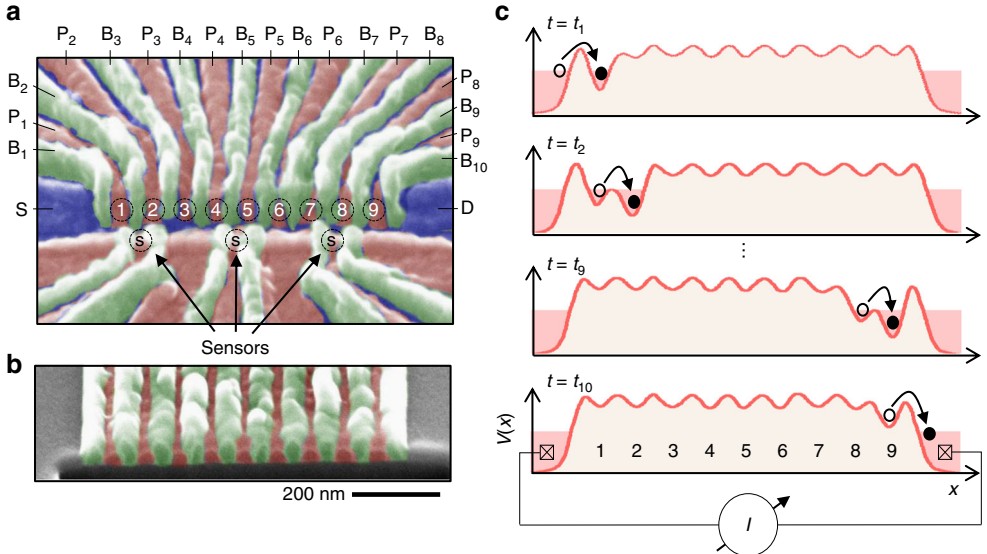

**Fig. 1** Si charge shuttle. **a** False-color SEM image of the device consisting of a 9-dot linear array (dots numbered from left to right) with 3 proximal charge sensors (denoted with a circled 'S'). The potentials of the dots are controlled by the plunger gates (pink) while the tunnel barriers are controlled by the barrier gates (green). The source and drain accumulation gate electrodes are shown in blue. **b** Tilted-angle cross-sectional SEM image of overlapping Al gates fabricated on a Si substrate. A focused ion beam was used to prepare the sample before imaging. **c** Illustration of the charge shuttling sequence showing the quantum dot confinement potential $V(x)$ at 4 different times during the shuttling sequence

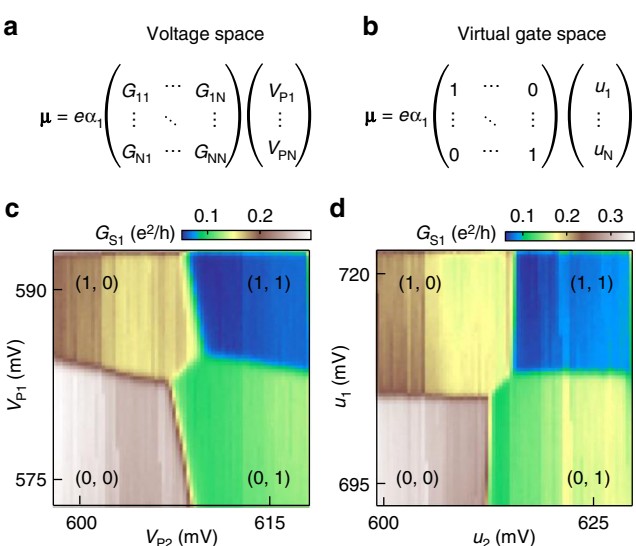

**Fig. 2** Defining a virtual gate voltage space. The chemical potential of the array expressed in gate voltage space **a**, and in virtual gate voltage space **b**. **c** DQD charge stability diagram for dots 1 and 2, measured as a function of the plunger gate voltages $V_{P1}$ and $V_{P2}$. **d** Charge stability diagram for the same DQD, but measured as a function of the virtual gate voltages $u_1$ and $u_2$.

coordinates, where a change in virtual gate voltage $u_1$ only shifts the chemical potential of dot 1 leaving the chemical potential of the dots in the remainder of the array unchanged. Extraction of the capacitance matrix from the data in Fig. 2c is detailed in the supplemental text (Supplementary Fig. 2). Figure 2d shows the charge stability diagram of the same DQD, but here plotted as a function of the virtual gate voltages $u_1$ and $u_2$. The dots 1 and 2 charge transitions are orthogonal in virtual gate voltage space, allowing for independent control of the chemical potential of each dot.

Larger few electron quantum dot arrays are built up by consecutively adding additional quantum dots to the right side of the device. With a DQD formed from dots 1 and 2, as shown in Fig. 2d, the interdot tunnel coupling $t_{c12}$ is tuned such that $t_{c12} \approx 5$ GHz $\approx 21$ μeV (Supplementary Fig. 4). Dot 3 is then tuned to the $N_3 = 0 \rightarrow 1$ charge transition, as verified in charge sensing. The formation of the third dot slightly affects the capacitance matrix for dots 1 and 2, requiring another calibration to establish the virtual gate voltage space $u_1$, $u_2$, and $u_3$. The tunnel coupling between dots 2 and 3 $t_{c23}$ is then tuned to $t_{c23} \approx 5$ GHz. Additional dots are added to the array following the same iterative tuning procedure and tunnel couplings are adjusted as necessary to maintain well-formed dots (Supplementary Fig. 3). To illustrate the formation of a 4 dot array, Figure 3a–c show pairwise charge stability diagrams that are plotted in virtual gate voltage space for dots 1 and 2 (Fig. 3a), dots 2 and 3 (Fig. 3b), and dots 3 and 4 (Fig. 3c). The remainder of the 9-dot array is configured by simply repeating this tune up procedure.

**Charge shuttling**. With a virtual gate voltage space established for the entire device, it is now possible to calculate a shuttling trajectory through the 9-dot charge stability space. For simplicity, the shuttling trajectory is outlined schematically in Fig. 3a–c for the 4-dot configuration (shuttling through the 9-dot array is demonstrated in Fig. 4). We initialize the system in the (0, 0, 0, 0) charge state by raising the chemical potentials of dots 1–4 above the Fermi level of the source and drain reservoirs. Here we extend the charge occupancy notation to ($N_1$, $N_2$, $N_3$, and $N_4$). We then increase $u_1$ within ~1 ns (step I in Fig. 3a) to transfer an electron from the source reservoir onto dot 1, with the device ending up deep in the (1, 0, 0, 0) regime. In step II of Fig. 3a, we move the electron across the (1, 0, 0, 0)–(0, 1, 0, 0) interdot charge transition. This interdot transition, and those that follow, must be performed adiabatically with respect to the interdot tunnel coupling in order to prevent charge shuttling errors from occurring (Supplementary Discussion). After the interdot charge transition is executed, we move the system deep into the (0, 1, 0, 0) charge

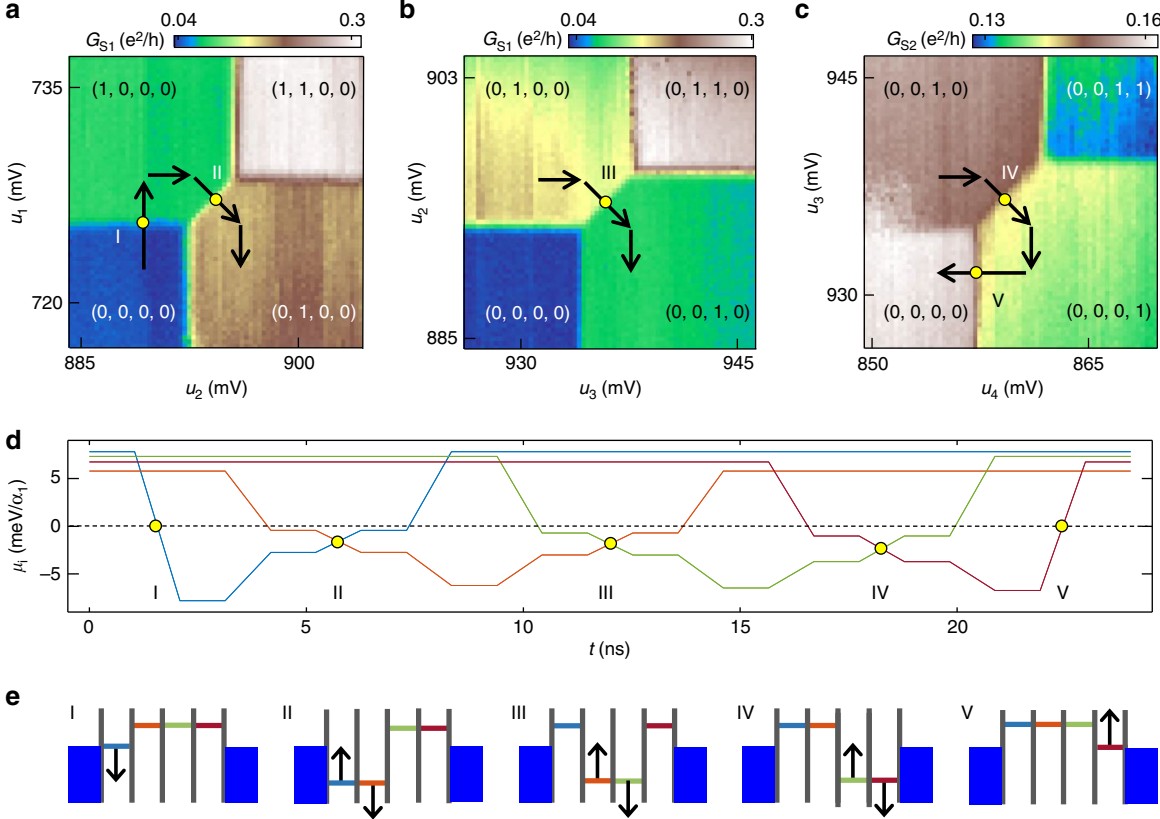

**Fig. 3** 4-dot charge shuttling trajectory **a**–**c**, Pairwise charge stability diagrams for dots 1–4, as measured in virtual gate voltage space. The leftmost charge sensor is used to acquire **a** and **b**, while the center charge sensor is used to acquire **c**. The shuttling trajectory through 4-dot charge stability space is overlaid on the data. Different portions of the pulse sequence are labeled with roman numerals for reference in **d** and **e**. **d** The chemical potential of dots (1, 2, 3, and 4) is plotted in (blue, orange, green, and maroon) for the shuttling sequence. The frequency of the shuttling sequence is varied by changing the length of the voltage plateaus, which alters the dwell time in each charge state. 1 ns ramp times between voltage plateaus are used in conjunction with low pass filters such that the interdot charge transfers are adiabatic with respect to interdot tunnel coupling. **e** Energy level diagrams at the positions in the pulse sequence marked with yellow dots in **a**–**d** and linked to the corresponding roman numerals. The arrows indicate the direction the energy levels are being swept and the colors of the levels correspond to the colors of the pulses in **d**

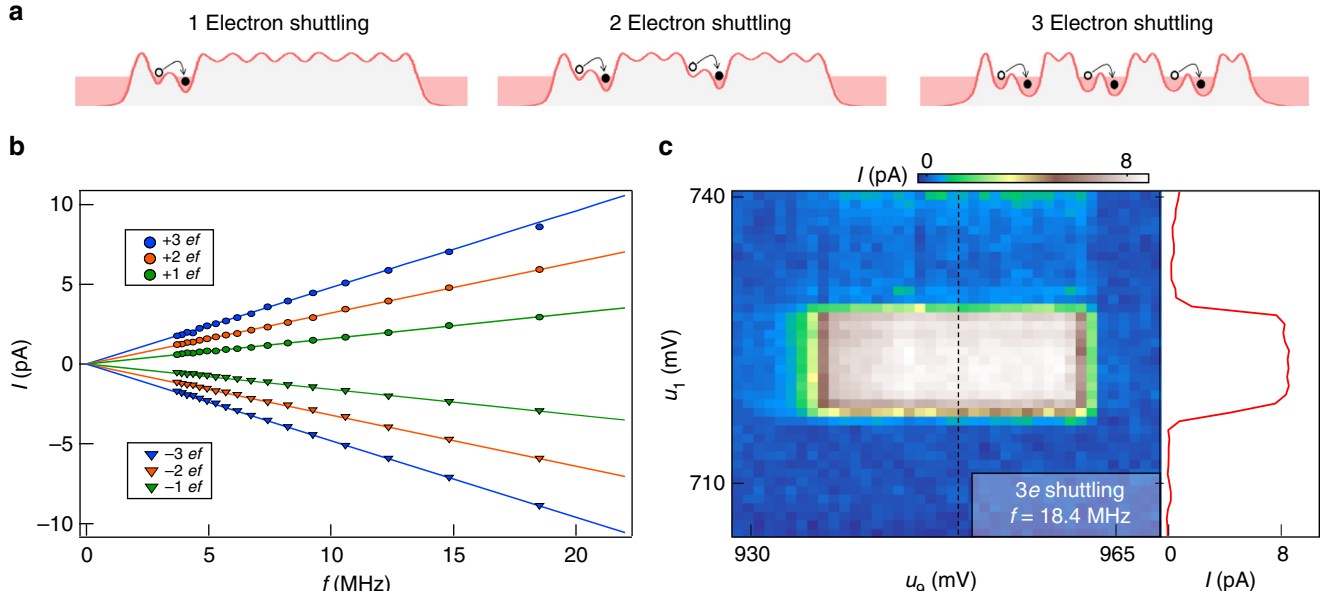

**Fig. 4** 9-dot charge shuttling. **a** Examples of single, double, and triple electron charge shuttling sequences. **b** Current $I$ measured as a function of frequency $f$ for single, double, and triple electron charge shuttling in both the forward and reverse directions. For reference, the solid line shows the expected current $I = nef$ for $-3 \le n \le 3$. **c** $I$ measured as a function of $u_1$ and $u_9$ during continuous 3-electron charge shuttling. A robust region of pumped current is observed. Right panel: a line-cut through the data shows a stable plateau of pumped current

regime with the chemical potential of dot 1 brought above the Fermi level of the source reservoir in order to ensure the electron only moves forward in subsequent portions of the shuttling sequence. The (0, 1, 0, 0)–(0, 0, 1, 0) and (0, 0, 1, 0)–(0, 0, 0, 1) interdot charge transitions are crossed in the same way (see steps III and IV in Fig. 3b–c), bringing the electron to dot 4. The final step in the pulse sequence (step V in Fig. 3c) transfers the electron from dot 4 to the drain reservoir and returns the device to the (0, 0, 0, 0) charge state.

It is helpful to visualize the charge shuttling sequence by examining how the chemical potential of each dot in the shuttle evolves in time. Figure 3d shows the chemical potential $\mu_i$ of each dot relative to the Fermi level of the source and drain electrodes as a function of time (The units of $\mu_i$ are meV/$\alpha_1$). The amplitude of the pulses varies from dot to dot due to slight variations in the charging energy across the array. Figure 3e shows energy level diagrams for the 4-dot system at five different instants of time, corresponding to the yellow dots in Fig. 3a–d. The black arrows in Fig. 3e indicate if the chemical potential is increasing or decreasing with time. Note that a conversion from $u_i$ to $V_{Pi}$ is required to program the pulse generator that is used for the shuttling sequence (Supplementary Discussion).

The four dot charge shuttling sequence described in Fig. 3 can be extended to the full 9-dot array by including pulses to execute the interdot transitions associated with dots 4–9. In principle, the pulse sequence can be used for QST of a spin qubit in even larger 1D arrays as long as the total shuttling time is less than $T_2^*$. To evaluate the performance of the charge shuttle, we measure the current $I$ pumped through the device as a function of the shuttling frequency $f$. To vary $f$, we change the dwell time in each charge state by altering the length of time at each voltage plateau in our pulse sequence (Fig. 3d), while keeping the ramp rates between voltage plateaus fixed. Figure 4b shows $I$ as a function of $f$ for the one electron charge shuttling sequence. The pumped current closely follows the expected $I = ef$ relation over the entire range of $f$ explored here. The shuttling direction can be changed by simply reversing the order of the pulse sequence, which yields $I = -ef$.

More complex shuttling trajectories that transfer 2 or 3 electrons across the device at a time can also be executed. These pulse sequences are schematically illustrated in Fig. 4a. For the 2 electron shuttling sequence, the electrons are separated by at least 3 empty dots. The middle panel of Fig. 4a illustrates the (1, 0, 0, 0, 1, 0, 0, 0, 0) → (0, 1, 0, 0, 0, 1, 0, 0, 0) charge transfer. In the 3 electron shuttling sequence, the electrons are separated by two dots, as illustrated in the right panel of Fig. 4a. The 2 and 3 electron shuttling sequences produce the expected $I = 2ef$ and $I = 3ef$ pumped currents (Fig. 4b). Moreover, as with the 1 electron shuttling sequence, these pulse sequences can be reversed, yielding $I = -2ef$ and $I = -3ef$.

To illustrate the robustness of the 9-dot charge shuttle, Fig. 4c shows the pumped current $I$ as a function of $u_1$ and $u_9$ that results from the 3 electron forward shuttling sequence. In contrast to conventional triple point charge pumping, where the pumped current can be a sensitive function of the gate voltages[40,41] we observe a broad plateau of pumped current due to the orthogonality of the virtual gate tuning parameters. The interior virtual gates ($u_2$–$u_8$) have similar 10–20 mV operating ranges determined by performing similar sweeps of virtual gate pairs within the array. The operating range is mostly influenced by the amplitude of the voltage pulses on each virtual gate (see Fig. 3d and Supplementary Discussion), and is limited by neighboring gate offsets such that the potential of dot $i$ must be below dot $i-1$ during charge transfer. While we do not claim that this device will be useful for metrology applications[36,42,43] due to the

small magnitude of the pumped current, the 2–3% errors that we observe in the highest pumped currents are entirely consistent with the 3% gain accuracy of the current amplifier used in these measurements. Errors due to cotunneling are predicted to be very small in multi-junction pumps[39]. A more precise characterization of the error rate could be performed using single charge detection[44].

In summary, we have shown that we can shuttle individual electrons across an extended 9-dot array using a bucket brigade approach at a rate that is more than three orders of magnitude faster than the inhomogeneous spin dephasing time in isotopically enriched Si. The shuttling sequence is easily parallelized to simultaneously move up to 3 electrons across the array. Our virtual gate approach for traversing the 9D charge stability space can be scaled to larger 1D quantum dot arrays, and may also be applicable to 2D arrays[35], making charge shuttling an attractive means to perform intermediate-scale QST within spin-based quantum processors. While this work has demonstrated shuttling of charges through a large ~1 µm 1D quantum dot array, it may be extended to examine spin shuttling in Si and the impacts of valley states[45], spin-relaxation "hot-spots"[46], and motional narrowing[32] on the spin transfer fidelity.

## Data Availability

Data available on request from the authors. The source data underlying Figs. 2c, d, 3a–c, 4b, c, and Supplementary Figs. 2, 3a–f, 4, 6, and Supplementary Table 1 are provided as a Source Data file.

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

## Acknowledgements

We thank Lisa Edge for providing the heterostructure used in these experiments and Felix Borjans for assistance with the focused ion beam system. Funded by Army Research Office grant W911NF-15-1-0149, the Gordon and Betty Moore Foundation's EPiQS Initiative through grant GBMF4535, with partial support from NSF grants DMR-1409556 and DMR-1420541. Devices were fabricated in the Princeton University Quantum Device Nanofabrication Laboratory.

## Author contributions

A.R.M and D.M.Z carried out the measurements with input from F.J.S. and J.R.P.; D.M.Z. fabricated the device. T.M.H. performed early measurements on a different device. M.J.G provided theory support. A.R.M., M.J.G. and J.R.P wrote the manuscript. All authors discussed the results and commented on the manuscript.

## Additional information

**Competing interests:** The authors declare no competing interests.

