## [Peer Review File · Nature Communications]

Reviewers' comments:

Reviewer #1 (Remarks to the Author):

Review of the manuscript "Shuttling a single charge across a one-dimensional array of silicon quantum dots" by A. R. Mills et al., submitted to Nature Communications.

In this work, the authors report on transferring a single (two and three) electron charge(s) across an array of nine quantum dots within 50 ns in a $^{28}\text{Si}/\text{SiGe}$ heterostructure. Such bucket brigade charge transfer were demonstrated in $\text{GaAs}/(\text{Al,Ga})\text{As}$ before, but never in such a long array of quantum dots. It requires a well-developed tuning scheme for the applied gate voltages. Furthermore, a single electron spin in isotopically purified Si/SiGe has excellent properties for defining a quantum bit, despite concerns about the crystal quality of Si/SiGe compared to $\text{GaAs}/(\text{Al,Ga})\text{As}$. The authors tackle one of the remaining challenges left towards a scalable quantum computer architecture: quantum state transfer over a length range of 50 nm to 10 microns. Their result is a very important step towards a multi-qubit silicon device. The authors demonstrate that the chemical potentials and tunnel couplings of the 9 quantum dot array can be sufficiently and fast controlled to render such a charge transfer possible. The result is timely and interesting for the readership of Nature Communications. The paper is very well written and the supplements explain very well the strategy for the voltage tuning of this complex device. I definitely recommend the publication in Nature Communications after considering these minor points:

1. In the introduction and conclusion the shuttling time is compared to the T_2^* spin dephasing time in natural Si of 1 μs . The authors use a rough and pessimistic estimate which actually underestimates the performance of their shuttle. The authors use an isotopically purified silicon device, thus the T_2^* for comparison should be longer. For natural silicon the elongation of the T_2^* time due to motional narrowing is not taken into account and could be mentioned in the introduction.
2. In Figure 3d just before and after Landau Zener transitions the control voltages are constant for some time. Is this "shuttle pause" used to control the frequency of the shuttle process or is it required for some other purpose? Please label it accordingly.
3. Figure 4c shows the robustness of the shuttle with respect to the virtual gates u_1 and u_9 . Could you please also list the possible voltage range for virtual gates u_2, \dots, u_8 for completeness. To give an impression of the homogeneity of the device, I would like to encourage the authors to give values for $V_{\text{PJ}}^{\text{off}}$, C and C_V within the supplements.
4. Page 5, 2nd paragraph "Figure 4c shows I as a function of...". Change to "Figure 4b shows...".

Reviewer #2 (Remarks to the Author):

The work by Mills et al describes the shuttelling of single electrons through a linear array of nine gate defined Si quantum dots (QD). The work is motivated by the future prospect of transferring spin information through the array by transferring spin-polarized charges.

For the controlled transfer of the charges through the linear QD array the capacitive coupling of all gates is taken into account by introducing the concept of a virtual gate space where the "virtual" gate voltages form an orthogonal space (i.e. they do not cross couple to neighbouring dots). Although this is a technical issue I think it is an interesting concept which will have future applications. The authors demonstrate pumping of up to three charges simultaneously through the linear array with pump frequencies above 10 MHz. The obtained error rates are of the order of a few percent where the uncertainty might already be dominated by the measurement electronics.

I think that the work is interesting for the community of spin qubits and possibly could find some interest in the field of electrical metrology. It should be published once the following remarks and comments are taken into account:

- What is the impact of cotunnelling on the error rate and the stability of pumping?
- What is the charging energy of the QDs? Is it the same for all dots?
- Do the authors expect the tuneable inter dot coupling to affect the spin dephasing time? This is a speculative question at present but it should be discussed with respect to the feasibility of the future application.
- How are the neighbouring gates insulated electrically? I suppose it is by an oxide on the Al surface but it should be noted somewhere to make the paper understandable without reading the prior work.
- The authors only cite very few references on the field of metrological single electron pumps although a wealth of papers have been published. They should add some references on the following: Si based QD pumps (pioneered by Fujiwara et al), linear pump arrays (e.g. the 7 junction pump studied by Martinis et al), and precision measurements of pumped currents. Note, that there are also review papers available providing an overview over the SET metrology field e.g. by Kaestner and Kashcheyevs and others.

Reviewer #3 (Remarks to the Author):

Report on "Shuttling a single charge across a one-dimensional array of silicon quantum dots" (NCOMMS-18-30506-T)

This manuscript demonstrates charge transfer along a linear chain of dots using a sequence of gate pulses with transfer being affected on a timescale eminently suitable for QST. A method to progressively tune interdot coupling is demonstrated with the concept of virtual gates simplifying the process of stepping through the charge stability diagrams.

The dot chain is gate defined in a Si/SiGe heterostructure and operated in accumulation mode. To shuttle electrons through the device the plunger of the first dot is lowered allowing one electron to tunnel in, the plungers on successive dots are modulated through three levels to shuffle the electron along. It appears that the barriers remain fixed during operation and Ref 15 explains how in plane confinement is achieved in the remaining perpendicular direction (Al screening gate.)

The need for, and concept of, virtual gates is explained such that the complicating effect of cross capacitance can be largely removed. The detail given in the main text and supplementary material is definitely sufficient for others to implement this concept. This idea of defining virtual parameters which can be swept or stepped is not completely new and simpler variations on this theme have been in use for some time (for example the detuning parameter in double quantum dots) and virtual gates are described and used in a 2017 paper from Vandersypen group [Hensgens et al. Quantum simulation of a Fermi-Hubbard model using a semiconductor quantum dot array Nature 548, 70] however this manuscript is an important addition and has a more thorough explanation of how to implement this idea than the above paper. The ideas in this manuscript are relevant to the entire gate defined semiconductor dot community and very likely to be of interest.

In the supplementary material the authors point out, reassuringly, that only a limited number of neighboring gates (3rd order) need to be considered when determining cross capacitance matrices for these sort of devices.

A detailed description of device operation using the virtual gates across dots 1-4 is given with additional pumping procedures for moving more closely spaced electrons.

A description of $n=3$ pumping as an illustration of shuttle robustness is provided. I am not sure that it is completely clear from the text and Figs 4b&c & Fig S4(inset) that the shuttle procedure is as robust as this data initially suggests. If we take the case of $n=1$ pumping a failure of the one transfer (say from 5 to 6) may well be masked by the next cycle where a second electron comes along to 5 and then from 5 through to 9 and the drain two electrons are being shuttled. This would

count as a QST failure although it would not appear as a deviation in the $I = \text{nef}$ current. It may be that the authors have other pulse sequences showing this is not the case? For example, if following each forward 1-to-9 shuttle cycle a reverse 8-to-1 shuttle is run this would ensure a clear channel. If such a procedure did not alter the expected $I = \text{nef}$ current, you can be sure that no electrons are being left behind and then caught by the next shuttle cycle.

Further comments

The methods and data appear thorough and an appropriate level of detail is provided to understand and reproduce the work described. The supplementary material is detailed and appropriate to the main text. With the possible exception of the $I = \text{nef}$ robustness demonstration all conclusions seem well supported by the data and explanations.

Suggested improvements

* Coherence of the spin state during shuttling is a clear go/no-go decision point for the intermediate QST described in this work, a discussion of possibly error mechanisms (or why such errors are unlikely to occur and/or be a problem) would strengthen the paper in the robustness of pumping section.

* Fig S4: I was not clear if the 6% error is referring to the deviation at around $f = 5\text{MHz}$ or 45MHz , or elsewhere? Also the Fig would benefit from having a 3% preamp error bar indication, or whatever the total error estimate.

(* optional) Fig 4c shows a stable plateau for u_1 vs u_9 variation. It would be interesting to know how much the length of this plateau varies with u_1 vs u_2 , u_1 vs u_3 etc. My expectation would be that u_1 and u_9 have the narrowest operating ranges and a comment on this in the paper or supplementary material would be nice – but is not required.

References:

* It would be appropriate to reference the work of Hensgens et al [Nat 548, 70] regarding virtual gates.

The authors make no claim about automating their setup procedure but if they wish to do so then Baart [APL 108, 213104] may be an appropriate reference for this.

And now taking my reviewer hat off... Were I to attend a talk on this topic, and I would certainly enjoy doing so, how plateau length varies with u_3 , u_3 etc is one of the questions I would ask. Also the same sort of data from Fig4c but taken for real gates ranges P_1 vs P_2 etc would make an interesting contrast with the u_1 vs u_2 picture and nicely show the power of virtual gates which I am sure will quickly become an "industry standard" technique.

End of Report

Response to Referee #1

Referee #1 shows strong support for the publication of our work, stating “The paper is very well written and the supplements explain very well the strategy for the voltage tuning of this complex device. I definitely recommend the publication in Nature Communications after considering these minor points.” The referee has several comments on the manuscript, which we address below.

Referee Comment #1: “In the introduction and conclusion the shuttling time is compared to the T_2^* spin dephasing time in natural Si of 1 μs . The authors use a rough and pessimistic estimate which actually underestimates the performance of their shuttle. The authors use an isotopically purified silicon device, thus the T_2^* for comparison should be longer. For natural silicon the elongation of the T_2^* time due to motional narrowing is not taken into account and could be mentioned in the introduction.”

Response: It is true that T_2^* will be significantly longer in ^{28}Si devices and we may also benefit from motional narrowing during the spin shuttling process. We have updated the manuscript to include the ^{28}Si T_2^* . We now mention motional narrowing in the introduction and cite related work on the topic (Flentje *et al.* 2017). We choose to keep the discussion of spin physics short in this manuscript since our data only demonstrate charge shuttling in a 9 dot array.

Referee Comment #2: “In Figure 3d just before and after Landau Zener transitions the control voltages are constant for some time. Is this “shuttle pause” used to control the frequency of the shuttle process or is it required for some other purpose? Please label it accordingly.”

Response: We have added clarification to the figure caption, explaining that the periods of constant voltage are used to vary the overall frequency of the shuttle, while keeping the interdot transition rate fixed. We also improved the wording on page 5 related to this point.

Referee Comment #3: “Figure 4c shows the robustness of the shuttle with respect to the virtual gates u_1 and u_9 . Could you please also list the possible voltage range for virtual gates u_2, \dots, u_8 for completeness. To give an impression of the homogeneity of the device, I would like to encourage the authors to give values for $V_{\text{PJ}}^{\text{off}}$, C and C_V within the supplements.”

Response: Referee 3 also suggested including more information in the manuscript. The requested values are included in the last two pages of the supplementary material.

When the array is tuned to the correct voltage offsets each virtual gate can be moved by 10 – 20 mV and the shuttling process is rather robust. We added the approximate operating ranges to the text and explained how these values are related to the amplitudes of the pulses, which are themselves calculated from the charging energies. One can then infer from the charging energies the approximate operating ranges for each dot in the array.

Referee Comment # 4: “Page 5, 2nd paragraph “Figure 4c shows I as a function of...”. Change to “Figure 4b shows...””

Response: We thank the Referee for carefully reading the manuscript and now cite the correct figure panel.

Response to Referee #2

Referee #2 shows similar support for the work and suggests it may also garner interest in the electrical metrology community. We have used the feedback from Referee #2 to improve our manuscript and supplementary material. We have also added more citations to relevant metrological work.

Referee Comment #1: “What is the impact of cotunnelling on the error rate and the stability of pumping?”

Response: Based on work by Jensen and Martinis [Accuracy of the electron pump, Phys. Rev. B **46**, 13407 (1992)] and the discussion by Keller in [Eur. Phys. J. Special Topics **172**, 297-309 (2009)] we would expect the error rate due to cotunneling, ϵ_{ct} , to decrease with the number of dots, N , as

$$\epsilon_{ct} \propto \left(\frac{R_K}{R_T}\right)^{N-1} \left(\frac{\delta E}{\Delta E}\right)^{2N-1}.$$

Here R_K is the resistance quantum h/e^2 and R_T is the junction resistance. The energy terms δE and ΔE are the energy range of available final states and the height of the barrier between the two ends of the pump. Keller estimates that a system with 5 or more junctions would have an error rate per cycle less than 10^{-8} . With our noise floor of about 100 fA, we would only expect to be able to reliably measure errors that occur at a rate greater than 10^{-4} per cycle. Based on the analysis in Keller 2009, our error rate due to cotunneling should be below our ability to measure.

We compute our pulses such that the amplitudes are only about half of the charging energy for each dot. This pulse amplitude keeps us from accidentally shuttling two electrons in a single dot, as the dot remains well away from loading an extra electron at its lowest potential.

Referee Comment #2: “What is the charging energy of the QDs? Is it the same for all dots?”

Response: We now include a table of charging energies in the supplement. The charging energies are fairly uniform across the array.

Referee Comment #3: “Do the authors expect the tuneable inter dot coupling to affect the spin dephasing time? This is a speculative question at present but it should be discussed with respect to the feasibility of the future application.”

Response: The interdot tunnel coupling is not varied during a shuttling sequence. We only vary the tunnel coupling during the device tune up, such as to make the interdot charge transitions adiabatic with respect to tunnel coupling. It is known from past experiments that tunnel coupling and Zeeman splitting can lead to spin relaxation hotspots. A reference to related spin relaxation physics is now given [Srinivasa *et al.*, “Simultaneous spin-charge relaxation in double quantum dots,” Phys. Rev. Lett. **112**, 226803 (2014)]. We added a statement to the conclusion of the manuscript indicating that spin relaxation hotspots will have to be taken into account during the spin shuttling sequence.

Referee Comment #4: “How are the neighbouring gates insulated electrically? I suppose it is by an oxide on the Al surface but it should be noted somewhere to make the paper understandable without reading the prior work.”

Response: The gate layers are insulated using native aluminum oxide. As suggested, we have added this detail to the paragraph describing the device fabrication.

Referee Comment #5: “The authors only cite very few references on the field of metrological single electron pumps although a wealth of papers have been published. They should add some references on the following: Si based QD pumps (pioneered by Fujiwara et al), linear pump arrays (e.g. the 7 junction pump studied by Martinis et al), and precision measurements of pumped currents.”

Response: When writing the manuscript, we did not want to make a claim that our device would be useful for metrological purposes (it is likely too slow). As such, we did not cite many metrology papers. The referee points out that even though this isn't a metrology paper, single electron pumps have been widely studied before and the some of the work should be cited. We agree and have added citations to Fujiwara's work, as well as Martinis' 7 junction pump and Mottonen's work on pumping in silicon.

Response to Referee #3

The Referee is positive about our work, stating “The ideas in this manuscript are relevant to the entire gate defined semiconductor dot community and very likely to be of interest.” The Referee's comments helped us to improve the manuscript and clarify some of the finer points related to the stability and expected behavior of the shuttle.

Referee Comment #1: “A description of $n=3$ pumping as an illustration of shuttle robustness is provided. I am not sure that it is completely clear from the text and Figs 4b&c & Fig S4(inset) that the shuttle procedure is as robust as this data initially suggests. If we take the case of $n=1$ pumping a failure of the one transfer (say from 5 to 6) may well be masked by the next cycle where a second electron comes along to 5 and then from 5 through to 9 and the drain two electrons are being shuttled. This would count as a QST failure although it would not appear as a deviation in the $I = nef$ current. It may be that the authors have other pulse sequences showing this is not the case? For example, if following each forward 1-to-9 shuttle cycle a reverse 8-to-1 shuttle is run this would ensure a clear channel. If such a procedure did not alter the expected $I = nef$ current, you can be sure that no electrons are being left behind and then caught by the next shuttle cycle.”

Response: The Referee points out that there may be some shuttle errors that are masked by our measurement of $I = ef$. The main source of error feasible in the shuttling sequence is failure to transfer an electron to the next dot in the array. In this case, could the next pulse sequence carry two electrons to the end of the array, effectively masking the error in the pumped current? One precaution we took against mistakenly shuttling two electrons in the same dot is computing the amplitudes of the pulses to be approximately $\frac{1}{2}$ of the charging energies of the dots. This pulse setting, along with setting the initial voltage offsets above the Fermi sea, means that at the lowest potential during 'dwell' the dots are still far away from adding another electron.

In the case of an electron getting stuck in dot i , what should happen with highest probability is the next electron is shuttled to dot $i-1$ where it is then blocked from moving to dot i . (The energy cost of adding a second electron is going to be an additional ~ 2 meV for the average dot, corresponding to an extra ~ 10 mV pulse on the plunger gates.) The shuttle sequence should then carry the initial electron stuck in dot i to the lead, effectively transferring the error backwards one dot. After i shuttle cycles, the error should propagate back to the lead, reinitializing the array in the $(0,0,\dots,0)$ configuration. If this were to happen often enough, we would expect a measurable decrease in current. To test for this error mechanism, we purposely reduced the interdot tunnel coupling, as illustrated in the supplementary Fig. S4. These data show that a measurement of $I = ef$ is indeed sensitive to the error mechanism raised by the referee.

To clarify this point, we have added a statement regarding the computation of the amplitudes of the pulses, and the physics behind this choice. We believe that this statement, combined with the text on “Probing errors from non-adiabatic dynamics” help clarify that we wouldn’t expect a shuttle with a current so close to $I = ef$ to have errors occurring frequently. We agree with the Referee that spin shuttling will be more susceptible to errors (relaxation, decoherence, valleys...). We do not claim that robust quantum state transfer has been achieved, only that robust $I = nef$ charge shuttling is possible in our device platform. Future experiments will investigate spin shuttling and associated errors.

Referee Comment #2: (Suggested improvement) “Coherence of the spin state during shuttling is a clear go/no-go decision point for the intermediate QST described in this work, a discussion of possibly error mechanisms (or why such errors are unlikely to occur and/or be a problem) would strengthen the paper in the robustness of pumping section.”

Response: We agree with the referee that this will be a clear decision point for implementing QST that we hope to extend this work to. Although we indicate our aim is to implement a scalable spin shuttle for QST in the paper, we choose to keep our discussion of spin physics brief as we have no spin shuttling data in the manuscript, and instead focus on the charge shuttling techniques which could be beneficial to the quantum dot community on their own. We have improved our discussion in relation to charge pumping errors per the previous comment.

Referee Comment #3: “Fig S4: I was not clear if the 6% error is referring to the deviation at around $f=5\text{MHz}$ or 45MHz , or elsewhere? Also the Fig would benefit from having a 3% preamp error bar indication, or whatever the total error estimate.”

Response: We have clarified that our error estimate at the lower tunnel coupling value of 2.4 GHz comes from current shuttled at a frequency of 45.5 MHz. We have also included a 3% gain error bar in the inset of Fig. S4 per the suggestion.

Referee Comment #4: “(* optional) Fig 4c shows a stable plateau for u_1 vs u_9 variation. It would be interesting to know how much the length of this plateau varies with u_1 vs u_2 , u_1 vs u_3 etc. My expectation would be that u_1 and u_9 have the narrowest operating ranges and a comment on this in the paper or supplementary material would be nice – but is not required.”

Response: The Referee’s comment is similar to comment #3 from the first Referee. It is helpful to clarify the size of the operating ranges in virtual gate voltage space. We have added to the text a discussion on the operating ranges of the interior gates. When the array is tuned to all of the correct offset voltages, each virtual gate can be moved around 10 – 20 mV before moving the shuttle out of the operating range, as now mentioned in the manuscript. The operating range is set by the amplitudes of the pulses (as discussed above). When we vary two interior gates at a time, we observe right-triangular-shaped regions of pumped current in virtual gate space with the legs about 10 – 20 mV each. These regions are similar to the finite bias triangles, and in fact point in the opposite direction when the current is reversed. The shape comes from the relative positions of the energy levels during the interdot charge transfer. If dot i is too low in potential relative to dot $i+1$, then no charge transfer occurs.

Referee Comment #5: “It would be appropriate to reference the work of Hensgens et al [Nat 548, 70] regarding virtual gates.”

Response: We have added a reference to Hensgens in the main text as well as references to early work using virtual gates.

Summary of changes to the manuscript

1. Introduction, pg. 2: Changed comparison of the shuttle time to T_2^* in isotopically purified silicon and changed reference to Veldhorst *et al.* (2014).
2. Introduction, pg. 2: Added mention of the potential impact of motional narrowing on the dephasing time during shuttling to the introduction and cited Flentje *et al.* (2017).
3. Second paragraph, pg. 2: Clarified how the overlapping Al gate-electrodes are insulated.
4. Page 3, 2nd paragraph: Added mention to previous work with virtual gates including citations of Keller *et al.* (1996), Nowack *et al.* (2011), and Hensgens *et al.* (2017).
5. Page 3, last paragraph: The lever-arm is given.
6. Page 5, second paragraph: Changed “magnitude” to “amplitude” to match the discussion in the supplemental text.
7. Page 5, third paragraph: Clarified that the plateaus in the pulse sequence are used for controlling the pulse frequency in this work in response to Reviewer #1’s comment #2.
8. Page 5, third paragraph: Corrected typo, changing “Figure 4c” to “Figure 4b”
9. On page 6, we added the approximate operating ranges of the interior virtual gates during shuttling and stated the main influence on these values are the amplitudes of the pulses.
10. On page 6, we added discussion of expected error rates due to cotunneling in multi-junction pumps and cited Martinis *et al.* (1994).
11. On page 6, after “metrology applications” we add citations to metrology applications [Yamahata *et al.* (2017), Rossi *et al.* (2014), and Keller *et al.* (1996)]
12. Conclusion: In the outlook portion, where we mention spin physics that we would like to investigate in the future, we have added discussion of the possible effects of motional narrowing and spin-relaxation hot-spots that could influence our shuttle. With these closing remarks, we have referenced a paper on spin relaxation due to valley states [Yang *et al.* (2013)], spin-relaxation at finite detuning [Srinivasa *et al.* (2013)], and motional narrowing [Flentje *et al.* (2017)].
13. Fig. 3 caption: Added to text to better explain Fig. 3d and clarify what the voltage plateaus are for.
14. SOM end of Section S1: Added a paragraph regarding the characteristics of the device (charging energies, offsets, capacitance matrices).
15. SOM in Section S1: Referenced the added table of offset voltages and charging energies.
16. SOM end of Section S2, added a statement on how the amplitudes of the pulses are computed and the physical motivation behind the computation.
17. SOM Fig. S4 caption: Clarified that the 6% error measured with $t_c = 2.4$ GHz was when pumping at 45.5 MHz.
18. SOM Fig S4 caption: Added description of the grey 3% gain error bar added to the inset figure.
19. SOM Fig S4 inset: Added 3% gain error bar on ideal current measurement in Fig. S4.
20. SOM Fig S5: Added the matrices C , C_v , R and v to the SOM.

21. SOM Table S1: Added a table of the addition energies and gate voltage offsets for each dot to demonstrate the homogeneity of the device.

REVIEWERS' COMMENTS:

Reviewer #3 (Remarks to the Author):

The authors have addressed my questions and I am fully content with their responses. There are two optional suggestions below but regardless of this I am happy to recommend publication of this interesting work.

Page 2 (optional)

We also note that spin dephasing due to hyperfine coupling in natural silicon may be suppressed by motional narrowing during the shuttling process, making the shuttling approach applicable to a variety of host materials [32]

The intention is clear, but this sentence potentially suggests that the possibility of motional narrowing in Si is what enables this approach in other materials. However, I am not an editor/sub-editor.

[32] mentions Si, SiGe and GaAs so "variety of host materials" is properly backed up.

PAGE 6 (optional)

"The interior virtual gates (u2-u8) have..." Interior virtual gates is not a previously defined term so it may be clearer to add the parenthesis (u2-u8).